# Assessment of PSMA Expression of Healthy Organs in Different Stages of Prostate Cancer Using [^68^Ga]Ga-PSMA-11-PET Examinations

**DOI:** 10.3390/cancers16081514

**Published:** 2024-04-16

**Authors:** Holger Einspieler, Kilian Kluge, David Haberl, Katrin Schatz, Lukas Nics, Stefan Schmitl, Barbara Katharina Geist, Clemens P. Spielvogel, Bernhard Grubmüller, Pascal A. T. Baltzer, Gero Kramer, Shahrokh F. Shariat, Marcus Hacker, Sazan Rasul

**Affiliations:** 1Department of Biomedical Imaging and Image-Guided Therapy, Division of Nuclear Medicine, Medical University of Vienna, 1090 Vienna, Austria; holger.einspieler@meduniwien.ac.at (H.E.); kilian.kluge@meduniwien.ac.at (K.K.); david.haberl@meduniwien.ac.at (D.H.); n11712573@students.meduniwien.ac.at (K.S.); lukas.nics@meduniwien.ac.at (L.N.); stefan.schmitl@akhwien.at (S.S.); barbara.geist@meduniwien.ac.at (B.K.G.); clemens.spielvogel@meduniwien.ac.at (C.P.S.); marcus.hacker@meduniwien.ac.at (M.H.); 2Department of Urology and Andrology, University Hospital Krems, Karl Landsteiner University of Health Sciences, 3500 Krems, Austria; bernhardandreas.grubmueller@krems.lknoe.at; 3Department of Biomedical Imaging and Image-Guided Therapy, Division of General and Pediatric Radiology, Medical University of Vienna, 1090 Vienna, Austria; 4Department of Urology, Comprehensive Cancer Center, Vienna General Hospital, Medical University of Vienna, 1090 Vienna, Austria; gero.kramer@meduniwien.ac.at (G.K.); shahrokh.shariat@meduniwien.ac.at (S.F.S.); 5Department of Urology, Weill Cornell Medical College, New York, NY 10065, USA; 6Department of Urology, Second Faculty of Medicine, Charles University, 15006 Prague, Czech Republic; 7Department of Urology, University of Texas Southwestern Medical Center, Dallas, TX 75390, USA; 8Division of Urology, Department of Special Surgery, Jordan University Hospital, The University of Jordan, Amman 11942, Jordan

**Keywords:** prostate cancer, PSMA expression, PSMA-PET scans, mCRPC, radioligand therapy

## Abstract

**Simple Summary:**

Radioligand therapies targeting prostate-specific membrane antigen (PSMA) receptors are currently being investigated in several ongoing trials for their application in early stages of prostate cancer (PCa). The aim of this study is to identify the PSMA-ligand binding in patients with early and advanced stages of PCa by measuring changes in the PSMA uptake in various organs, including non-physiologically PSMA-expressing organs, as differences in PSMA expression could lead to unpredictable radiotoxicity in early-stage PCa patients. In addition, total tumor volume (TTV) was determined to evaluate differences in PSMA-ligand uptake related to TTV in the investigated cohorts.

**Abstract:**

The efficacy of radioligand therapy (RLT) targeting prostate-specific membrane antigen (PSMA) is currently being investigated for its application in patients with early-stage prostate cancer (PCa). However, little is known about PSMA expression in healthy organs in this cohort. Collectively, 202 [^68^Ga]Ga-PSMA-11 positron emission tomography (PET) scans from 152 patients were studied. Of these, 102 PET scans were from patients with primary PCa and hormone-sensitive biochemically recurrent PCa and 50 PET scans were from patients with metastatic castration-resistant PCa (mCRPC) before and after three cycles of [^177^Lu]Lu-PSMA-RLT. PSMA-standardized uptake values (SUV) were measured in multiple organs and PSMA-total tumor volume (PSMA-TTV) was determined in all cohorts. The measured PET parameters of the different cohorts were normalized to the bloodpool and compared using t- or Mann–Whitney U tests. Patients with early-stage PCa had lower PSMA-TTVs (10.39 mL vs. 462.42 mL, *p* < 0.001) and showed different SUVs in the thyroid, submandibular glands, heart, liver, kidneys, intestine, testes and bone marrow compared to patients with advanced CRPC, with all tests showing *p* < 0.05. Despite the differences in the PSMA-TTV of patients with mCRPC before and after [^177^Lu]Lu-PSMA-RLT (462.42 mL vs. 276.29 mL, *p* = 0.023), no significant organ differences in PET parameters were detected. These suggest different degrees of PSMA-ligand binding among patients with different stages of PCa that could influence radiotoxicity during earlier stages of disease in different organs when PSMA-RLT is administered.

## 1. Introduction

Prostate cancer (PCa) is one of the most common cancers and the fifth leading cause of cancer-associated death worldwide [1]. Localized PCa is usually treated with curative intention by surgical prostatectomy or local radiation therapy with or without androgen deprivation therapy (ADT) [2].

Prostate-specific antigen (PSA), produced exclusively by prostate epithelial cells, represents a predictor for PCa and offers a crucial marker for detecting tumor recurrence following initial therapy for PCa [3]. Nevertheless, 20–50% of PCa patients show an increase in serum PSA in the form of a biochemical recurrence (BCR) within 10 years of primary definite therapy [4,5]. At this clinical stage of the disease, ADT can be considered the treatment of choice for most patients among several treatment options [6]. ADT affects the secretion of hormones, including luteinizing hormone-releasing hormone, leading to a decrease in the production of androgens within the testes as well as in levels of circulating androgens in the body [6]. Owing to an array of mechanisms, including androgen receptor amplifications and mutations, ADT eventually leads to castration resistance, characterized by disease progression after 2–3 years despite castrated testosterone levels. [7]. Castration-resistant PCa (CRPC) accounts for a significant proportion of prostate cancer cases and develops in approximately 20% of men in follow-up care. In over 80% of them, the disease progresses and reaches a more aggressive and advanced stage of disease in terms of metastatic CRPC (mCRPC) [8].

In this regard, prostate-specific membrane antigen (PSMA), also known as glutamate-carboxypeptidase II, is an enzyme and receptor that is increasingly expressed on the surface of prostate cells of patients with mCRPC [9]. It is an attractive target for the diagnosis and treatment of PCa and, thus, molecular imaging with positron emission tomography (PET) targeting these receptors is now increasingly used worldwide for the staging and follow-up of PCa patients. This examination has already yielded higher diagnostic accuracy and lower radiation exposure in patients with metastatic PCa than conventional imaging procedures such as computed tomography (CT) and bone scans [10]. Even though the diagnostic precision of PSMA-PET imaging increases with higher PSA levels, PSMA-PET scans may still be useful in PCa patients with a serum PSA < 0.5 ng/mL, with a sensitivity of 50% [11].

By selectively targeting PSMA-expressing tumor cells, radioligand therapies (RLT) like [^177^Lu]Lu-PSMA therapy have demonstrated promising results such as delayed disease progression and improved disease outcomes in patients with mCRPC. The therapy not only reduces PSA levels and prolongs radiographic progression-free survival, but is also associated with lower levels of toxicity compared to chemotherapy, thereby supporting its potential as an alternative therapy [12,13,14].

As a result, PSMA-RLT in earlier stages of PCa is currently being investigated in several ongoing studies, although its potential impact on this patient group is not sufficiently studied. Hence, the objective of this study is to provide an insight into the changes in PSMA uptake using [^68^Ga]Ga-PSMA-11 PET scans in various organs in patients with early and advanced stages of PCa. Moreover, we aimed to reveal the influence of PSMA-total tumor volume (PSMA-TTV) on PSMA’s organ-expression pattern among the cohort, as differences in this pattern could have major impacts on organ radiotoxicity in early-stage PCa patients.

## 2. Patients and Methods

### 2.1. Patients

This retrospective study included data from 152 patients with PCa. In total, [^68^Ga]Ga-PSMA-11 PET-CT and [^68^Ga]Ga-PSMA-11 PET-MRI (magnetic resonance imaging) scans of 102 patients with primary PCa and hormone-sensitive (HS) BCR and 50 patients with mCRPC before and after 3 cycles of [^177^Lu]Lu-PSMA-RLT were analyzed. All PET scans were performed between December 2014 and February 2020. This study was approved by the ethics committee of the Medical University of Vienna (EK: 1745/2021) and, due to its retrospective design, written informed consent was not required from patients for data collection and analysis. All studied patients with primary PCa exclusively received [^68^Ga]Ga-PSMA-11 PET-MRI (including multiparametric MRI) scans for initial tumor staging and detailed characterization of their primary lesions in the prostate. Moreover, patients with BCR received a [^68^Ga]Ga-PSMA-11 PET-CT scan following the latest guidelines of the European Association of Urology [15]. For mCRPC patients, the decision whether to perform a PET-CT or PET-MRI examination depended mainly on the availability of imaging slots for the corresponding imaging modality.

### 2.2. PET-MRI Examination

All PET-MRI scans were performed with a Biograph mMR system (Siemens, Erlangen, Germany), which consists of an MRI-compatible PET detector integrated with a 3.0-T whole-body MRI scanner. The PET component provided an axial field of view (FoV) of approximately 26 cm and a transverse FoV of 59 cm with a sensitivity of 13.2 cps/kBq, which was obtained utilizing a three-dimensional (3D) acquisition technique. Patients were administered an intravenous injection of 2 MBq/kg body weight [^68^Ga]Ga-PSMA-11 60 min prior to the start of the PET-MRI acquisition. A partial body PET scan was obtained from skull base to thighs with four bed positions (5 min sinogram mode each). PET images were reconstructed using 3 iterations and 21 subsets. Dixon-VIBE sequences were used for attenuation correction based on MRI, which included in-phase, opposed-phase, and fat-saturated as well as water-saturated images. For patients with BCR and mCRPC, the exact MRI sequences within these integrated PET-MRI scans were mentioned in a previously published study by Grubmüller et al. [16]. For patients with primary PCa, the detailed MRI sequences were reported in another previously published study [17].

### 2.3. PET-CT Examination

The PET-CT scans were acquired with a Siemens Biograph TruePoint PET-CT scanner (Siemens Healthineers, Erlangen, Germany). Whole-body scans from the skull base to the thigh were obtained 60 min after the tracer injection of [^68^Ga]Ga-PSMA-11, 2 MBq/kg body weight. CT scans were acquired at 120 kV and 230 mAs with a CT matrix size of 512 × 512. Contrast medium was administered unless there were contraindications for contrast application. PET examinations were performed with a duration of 4 min per bed position, with iterative reconstruction utilizing a point-spread-function-based algorithm. Afterwards, corrections for scatter and attenuation were applied based on the CT scan (PET matrix size, 168 × 168).

### 2.4. Image Analysis

PET image intensities (Bq/mL) were converted to standardized uptake values (SUV), which were normalized to the body weight. Two experienced nuclear medicine physicians determined the SUVmean and SUVmax values in the organs under the guidance of another nuclear medicine specialist with more than 10 years of experience in this field. They manually placed a cuboidal volume of interest (VOI) with a target size of 13.50 mL or 1.60 mL in the heart (left ventricle), liver (right lobe), pancreas (head region), spleen, small intestine (duodenum), bone marrow (left femoral head), muscle (left gluteus maximus), thyroid gland, lungs, kidneys (renal cortex), testes and both sides of the mandibular glands thereby avoiding the inclusion of areas with suspicious lesions or large vessels on CT or MRI. Afterwards, PET parameters were normalized to the blood pool by dividing the SUVmean and SUVmax of each organ by the SUVmean of the cuboid VOI placed into the abdominal aorta.

Image analysis was performed using a dedicated workstation with Hybrid 3D software (version 4.17, Hermes Medical Solutions, Stockholm, Sweden). PSMA-expressing tumor regions were first segmented on PET images using in-house-developed, deep learning-based research prototype software and were consecutively validated by nuclear medicine physicians using Slicer3D software (version 4.11). The PSMA tumor volumes (PSMA-TTV) in all patients were then calculated by counting all delineated voxels and multiplying by the corresponding voxel size of the image.

### 2.5. Protocol of PSMA-RLT

The protocol of PSMA-RLT consisted of 3 cycles of an intravenous administration of approximately 7400 MBq of [^177^Lu]Lu-PSMA-617 (from in-house production) with an interval of 4 weeks between the cycles. For each cycle, patients were hospitalized for 48–72 h and received one liter of normal saline solution intravenously at 300 mL/h 30 min prior to the administration of this radiopharmaceutical therapy agent. All of these patients underwent [^68^Ga]Ga-PSMA-11 PET-CT or PET-MRI before the first cycle and one month after the 3rd cycle of the therapy.

### 2.6. Statistical Analysis

Statistical analysis was performed with EasyMedStat software (version 3.20.4, EasyMedStat, Paris, France). Numeric variables were expressed as mean (±standard deviation (SD)) and as median (minimum; maximum), and discrete outcomes were expressed as absolute frequencies (N). The group comparability of the different PCa cohorts was assessed by comparing the baseline demographic data. The normality and heteroskedasticity of continuous data were assessed and calculated using the Shapiro–Wilk test and the Levene test. The unpaired Student *t*-test, the Welch *t*-test and the Mann–Whitney U test were used according to data distributions to compare continuous outcomes. Discrete outcomes were accordingly compared with the chi-square test or Fisher’s exact test. All of the tests used were two-tailed. In addition, the differences in PSMA-TTV were assessed with the Kruskal–Wallis test as required, depending on the type of data distribution. If the null hypothesis was rejected, then post-hoc pairwise analyses were conducted using the Dunn–Bonferroni test. The alpha risk was set at 5% (α = 0.05) for all statistical analyses.

## 3. Results

Demographic characteristics of the different studied cohorts are demonstrated in Table 1. As can be seen in that table, the median age of the patients was similar in all cohorts, although the number of patients with a previous history of hormonal therapy and chemotherapy, as well as Xofigo^®^ (Ra-223) therapy, was clearly higher in patients with mCRPC than in patients with primary PCa and HS-BCR PCa. In addition, patients with mCRPC had noticeably more bone and visceral metastases than patients with primary PCa and HS-BCR PCa.

### 3.1. PSMA Uptake in Different Cohorts

The median SUVmean and SUVmax values of organs in patients with early stages of PCa (primary PCa and HS-BCR) and in patients with mCRPC before and after PSMA-RLT, normalized to the blood pool, are illustrated in Table 2.

As can be seen in Table 2, patients with earlier stages of PCa showed higher levels of PSMA expression in comparison to patients with CRPC before [^177^Lu]Lu-PSMA-617 treatment, with significantly different values for SUVmean and SUVmax in the thyroid (both *p* < 0.001), submandibular glands (*p* = 0.004; *p* = 0.013), heart (both *p* < 0.001), liver (both *p* < 0.001), pancreas (both *p* < 0.001), kidneys (both *p* < 0.001), small intestine (*p* = 0.006; *p* = 0.002), testes (both *p* < 0.001) and bone marrow (*p* = 0.002; *p* = 0.002). No significant difference was found for spleen (*p* = 0.074; *p* = 0.102), lungs (*p* = 0.097; *p* = 0.135) and muscle (*p* = 0.365; *p* = 0.102).

In patients with mCRPC before and after [^177^Lu]Lu-PSMA-617 therapy, no significant difference in SUVmean or SUVmax was found in the thyroid (*p* = 0.867; *p* = 0.729), submandibular glands (*p* = 0.354; *p* = 0.256), heart (*p* = 0.791; *p* = 0.347), lungs (*p* = 0.171; *p* = 0.111), liver (*p* = 0.309; *p* = 0.137), pancreas (*p* = 0.746; *p* = 0.391), spleen (*p* = 0.915; *p* = 0.915), kidneys (*p* = 0.699; *p* = 0.649), small intestine (*p* = 0.528; *p* = 0.292), testes (*p* = 0.487; *p* = 0.305), bone marrow (*p* = 0.756; *p*= 0.783) or muscle (*p* = 0.775; *p* = 0.166).

### 3.2. PSMA-TTV in Different Cohorts

As expected, patients with primary PCa or HS-BCR showed statistically significantly lower PSMA-TTVs than patients with mCRPC [^177^Lu]Lu-PSMA-617 therapy (*p* < 0.001; mean 10.39 mL, 462.42 mL, respectively). Furthermore, PSMA-TTVs were significantly higher in patients with mCRPC P therapy than in patients after radioligand therapy (mean 462.42 mL versus 276.29 mL, *p* = 0.023). The median values of PSMA-TTV in ml in patients with early-stage PCa, and in those with mCRPC before and after PSMA-RLT were 1.47 (IQR 4.99), 161.32 (IQR 405.91) and 51.26 (IQR 245.22), respectively (Figure 1).

## 4. Discussion

The latest guidelines of the European Association of Urology recommend [^177^Lu]-PSMA-617 RLT as an effective treatment option for pre-treated mCRPC patients with one or more metastatic lesions that express PSMA at high levels that exceed that of uptake in the liver on diagnostic PSMA-PET-CT scan. Due to its good tolerability, antitumor properties, good PSA response rates and lower side effects than chemotherapy and hormone therapy, PSMA-RLT could also be a promising treatment option for patients with PCa with a low tumor burden and who have not yet reached the castration-resistant stage. Several ongoing trials are, therefore, currently investigating the efficacy of this RLT in metastatic HS PCa, such as the Bullseye and UpfrontPSMA trials, or in combination with hormone- and chemotherapies in patients with mCRPC, as in the LuCAB and LuPARP trials [18,19,20,21]. In other studies, the effect of this therapy was also explored as a neoadjuvant therapy in patients with primary oligometastatic PCa with promising outcomes, e.g., in the LuTectomy and NALuPROST trials [22,23].

In this study, we evaluated PSMA uptake in [^68^Ga]Ga-PSMA-11 PET scans in organs exhibiting physiological PSMA expression as well as in non-physiological PSMA-expressing organs in male patients with different stages of PCa. Indeed, the radiation of the PSMA tracer is not limited to tissues that physiologically express PSMA but affects the entire body, including all other organs. Our results indicated that PSMA uptake in various organs, both with and without physiological PSMA expression, is significantly higher in patients with early-stage PCa comprising primary PCa and HS-BCR compared to patients with advanced mCRPC. In a study by Peters et al., the authors examined the absorbed-dose ratios in organs and lesions from [^68^Ga]Ga-PSMA-11 PET-CT 1 h after the injection of the PSMA tracer and from SPECT/CT, which was performed shortly after PSMA-RLT. They demonstrated that the PSMA uptake in PSMA-PET scans can be used to predict the absorbed radiation dose of [^177^Lu]Lu-PSMA-617 therapy to organs and to metastatic lesions [24]. This suggested that a completely different level of PSMA uptake between different organs, as found in our study, could be used as a surrogate for estimating the different absorbed doses of PSMA-RLT that might potentially lead to unexpected or more severe side effects.

From the findings of previous studies, it seems that there are some well-studied factors that could potentially influence the ability of PSMA-labeled isotopes to be distributed in the organs. Kluge et al. examined [^68^Ga]Ga-PSMA-11 PET scans in PCa with regard to the ADT status of the patients. They found that PSMA uptake in endogenous-PSMA-expressing organs such as the liver, kidneys, spleen and salivary glands was lower in patients who were on ADT than in patients who were not on ADT at the time of the scan. These results indicate a systemic modulation of PSMA expression by hormone therapy with regard to ADT [25]. Another contributing factor that potentially influences PSMA uptake in the organs is the so-called tumor-sink effect. Depending on the tumor load, the biodistribution of PSMA-ligands might change. Usually, a high tumor load leads to lower PSMA uptake in normal, non-tumoral organs [26,27]. In a study by Groener et al. comparing PSMA uptake in [^68^Ga]Ga-PSMA-11 PET scans in patients with metastatic PCa before and after PSMA-RLT, uptake in liver tissue and salivary glands was inversely related to tumor burden before and after PSMA-RLT [28]. In our study, patients with mCRPC before and after three cycles of [^177^Lu]Lu-PSMA-617 therapy had significantly different PSMA-TTVs, but it seems that the difference in PSMA-TTV was too marginal to result in a relevant change in PSMA uptake in one single organ. Nevertheless, a discrete trend towards increased absolute PET parameters in a few organs after RLT (Table 1) could be observed, but the impact seems to be negligible and not significant. Thus, our results suggest that a maintenance therapy dose can be provided to patients with mCRPC after three cycles of [^177^Lu]Lu-PSMA radioligand therapy despite them having reduced PSMA-TTVs.

The main limitation of this study is its retrospective design. In addition, even when thoroughly drawn, the application of cuboid VOIs to estimate the PSMA uptake in individual organs rather than a whole-organ VOI is greatly influenced by the individual investigator, as there is still no validated software that can be used for PSMA-PET examinations to automatically delineate entire individual organs. Although the majority of our studied mCRPC patients were examined with the same PET-CT or PET-MRI scanner before and after three PSMA-RLT cycles to assess their response to therapy, the bias in SUVs due to the use of different scanners should also be considered. Here, discrete differences in SUVs were detected at least between PET-CT and PET-MRI examinations using [^18^F]FDG [29,30]. Additionally, it should be noted that the choice of imaging modality could also lead to some differences in tumor identification. Some prior studies have revealed that a PSMA-PET-MRI scan is associated with a slightly higher sensitivity for the detection of PCa recurrence and metastatic lesions than a PSMA-PET-CT scan, which was mainly due to the complementary additional information provided by MRI [31,32].

Henceforth, results of large prospective studies are required to demonstrate the clinical safety and efficacy of [^177^Lu]Lu-PSMA-RLT in patients with early stages of PCa.

## 5. Conclusions

The results of this study point to a systemic variation in PSMA uptake in patients’ organs in relation to different stages of PCa disease, and this could have implications for radiotoxicity at earlier stages of disease in different organs when PSMA-RLT is administered. Nevertheless, prospective studies with large populations are necessary to estimate its clinical safety and effectiveness at this tumor stage. In addition, our data indicate that a maintenance dose can be provided to patients with mCRPC after several cycles of [^177^Lu]Lu-PSMA-ligand therapy despite them having reduced PSMA-TTVs.

## Figures and Tables

**Figure 1 cancers-16-01514-f001:**
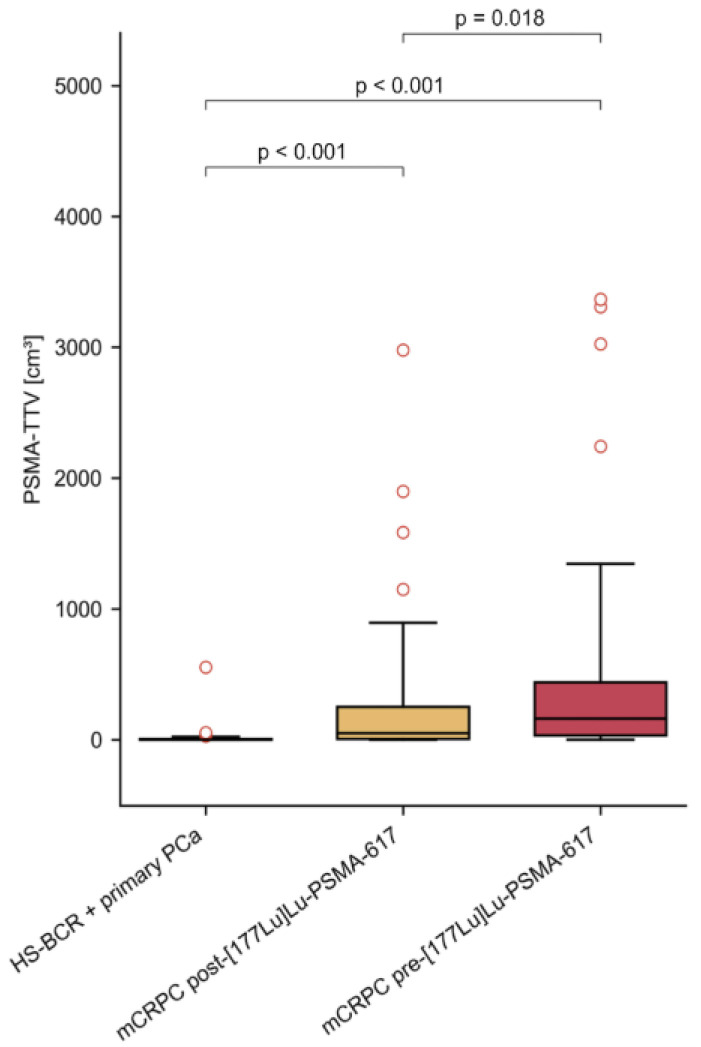
Distribution of PSMA-TTV of all studied cohorts with prostate cancer.

**Table 1 cancers-16-01514-t001:** Demographic characteristics of all studied cohorts with prostate cancer.

Parameters	Primary PCa + HS-BCR (N: 102)	mCRPC before PSMA-RLT (N: 50)	mCRPC after PSMA-RLT (N: 50)
Median Age in years (range)	71.5 (50; 87)	71.5 (51; 84)	72.0 (52; 84)
Previous therapies: Hormone therapy (N):	1	37	37
ADT (N)	1	28	28
Abiraterone/Enzalutamide (N)	0	32	32
Chemotherapy (N):	0	33	33
Docetaxel (N)	0	30	30
Docetaxel + Cabazitaxel (N)	0	19	19
Xofigo^®^ (Ra-223) (N)	0	15	15
Patients with metastatic lesions (N):	39	50	50
Lymph node (N)	33	38	37
Bone (N)	11	40	40
Visceral (N)	0	9	9
PET-CT (N)	31	10	3
PET-MRI (N)	71	40	47

N—Number of patients; PCa—Prostate cancer; HS-BCR—hormone-sensitive biochemical recurrent PCa; mCRPC—metastatic castration-resistant PCa; RLT—radioligand therapy, ADT—androgen deprivation therapy.

**Table 2 cancers-16-01514-t002:** PSMA-PET parameters of all studied cohorts with prostate cancer, normalized to the blood pool.

Organ	PSMA-PET Scans					
	Primary PCa + HS-BCR (N: 102)		mCRPC before PSMA-RLT (N: 50)		mCRPC after PSMA-RLT (N: 50)	
	SUVmean ^#^	SUVmax ^#^	SUVmean ^#^	SUVmax ^#^	SUVmean ^#^	SUVmax ^#^
Submandibular gland	17.53 * (1.94; 110.73)	24.33 * (2.49; 160.40)	12.07 * (3.88; 64.58)	17.93 * (5.14; 115.97)	10.25 (1.86; 31.15)	14.00 (2.89; 45.24)
Thyroid	1.70 * (0.18; 15.20)	2.70 * (0.23; 20.63)	1.15 * (0.43; 6.00)	1.92 * (0.67; 12.61)	1.18 (0.34; 3.06)	1.84 (0.49; 6.08)
Heart	1.10 * (0.36; 5.36)	2.22 * (0.81; 13.12)	0.81 * (0.27; 3.55)	1.57 * (0.51; 13.17)	0.84 (0.33; 2.81)	1.42 (0.58; 6.06)
Lung	0.47 (0.15; 2.53)	1.06 (0.35; 8.00)	0.38 (0.18; 3.42)	0.94 (0.41; 7.81)	0.38 (0.13; 4.83)	0.77 (0.22; 11.00)
Liver	5.74 * (1.77; 19.82)	8.09 * (2.63; 75.85)	3.59 * (1.19; 21.94)	5.43 * (1.97; 40.11)	3.00 (0.74; 9.14)	4.39 (1.48; 12.25)
Spleen	5.74 (1.25; 33.08)	10.05 (2.62; 164.56)	4.79 (1.11; 30.67)	7.37 (1.96; 73.17)	5.02 (1.06; 19.23)	7.70 (1.61; 41.78)
Kidney	38.09 * (2.03; 212.73)	55.38 * (2.91; 268.14)	18.99 * (3.75; 161.89)	29.50 * (4.84; 283.31)	19.38 (5.84; 69.07)	32.08 (8.88; 128.87)
Pancreas	2.44 * (0.66; 12.08)	4.00 * (1.03; 22.50)	1.71 * (0.54; 4.70)	2.48 * (0.68; 9.59)	1.56 (0.58; 3.78)	2.36 (0.82; 7.40)
Small intestine (duodenum)	2.63 * (0.10; 18.91)	4.30 * (0.47; 30.59)	1.84 * (0.27; 5.95)	3.38 * (0.84; 10.18)	1.50 (0.25; 10.14)	2.60 (0.64; 15.76)
Testes	1.58 * (0.39; 10.20)	2.65 * (0.84; 19.75)	0.80 * (0.16; 6.50)	1.40 * (0.37; 14.11)	0.70 (0.24; 14.89)	1.23 (0.46; 23.89)
Bone marrow	0.23 * (0.02; 1.29)	1.08 * (0.18; 7.52)	0.16 * (0.05; 0.99)	0.66 * (0.18; 6.65)	0.18 (0.05; 0.64)	0.65 (0.15; 3.15)
Muscle (gluteus maximus)	0.45 (0.16; 1.95)	1.04 (0.36; 4.97)	0.45 (0.18; 1.33)	0,97 (0.34; 5.39)	0.40 (0.16; 1.11)	0.81 (0.29; 2.40)

N—Number of patients; PCa—Prostate cancer; HS-BCR—hormone-sensitive biochemically recurrent PCa; mCRPC—metastatic castration-resistant PCa; RLT—radioligand therapy; ^#^—Values presented in median (minimum; maximum); *—Significant difference (*p* < 0.05) between patients with primary PCa + HS-BCR and patients with mCRPC before PSMA-RLT.

## Data Availability

The data can be shared up on request.

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
