# Peer review of "Assessment of PSMA Expression of Healthy Organs in Different Stages of Prostate Cancer Using [^68^Ga]Ga-PSMA-11-PET Examinations"

_cancers, 2024, doi:10.3390/cancers16081514_

Round 1

Reviewer 1 Report

Comments and Suggestions for Authors

This retrospective study is quite relevant and appropriate. The investigators aimed to study the influence of PSMA uptake on normal organs for prostate cancer patients. A second aim was to estimate the total tumor volume amongst the same cohort. The paper represents interesting findings, high quality figure/ tables nice layout and easy flow. However, the viewer has encountered some areas that can be improved before accepting this work for publication, including the methodology and discussion. Enclosed below are some general and specific comments for the authors to consider.

General comments

1. Write in short sentences to ease the flow and comprehension, example the abstract sentence line 32 to 36.     

2. Re-write your total sample size in a clearer format, it seems the total cohort included is 152 then breakdown.  

3. Patient undergo PET/CT vs PET/ MRI is determined based on what criteria? As this may impact tumor localization and size.  

4. Considerable level of uncertainty is included that has to do with exact placement of VOI. Was there any standardized or structured guide among the observer?

Specific comments:

1.Line 88: The first time the full radiopharmaceutical of [68Ga] Ga-PSMA-11 PET was introduced. Please clearly state or unify this throughout the manuscript whenever a PSMA PET was mentioned, it refers to the radiopharmaceutical.

2.Line 95: please clearly state if you are using a hybrid PET scanner whether with a CT or MRI and rewrite the sentence.

3. Image analysis section: how was the VOI placed on the fused image, manually or threshold based? Target delineation ROI/ VOI affects the mean SUV and size.

4. Line 150: to is missing after prior.

5. Line 171: as is missing after well.

Comments on the Quality of English Language

The manuscript is nicely written with only few edits required, highlighted in the specific comments.

Author Response

Please check the attached response. 

Reviewer 2 Report

Comments and Suggestions for Authors

This is a very interesting paper comparing the PSMA-targeted tracer uptake in normal tissue at different stages of prostate cancer with different tumor load, different therapies etc. Overall the presentation is very clear and the conclusions are supported by the data presented.

One question might be if more can be made of the Total tumor volume (TTV) data? Maybe a correlation test between TTV and uptake in other tissue, either for all patients or within the established groups? Intuitively it seems uptake in other tissue would be dependent on the number of PSMA receptors available in tumor tissue, so there should be a correlation?

Author Response

Please check the attached response. 
